# Polymer Cancerostatics Containing Cell-Penetrating Peptides: Internalization Efficacy Depends on Peptide Type and Spacer Length

**DOI:** 10.3390/pharmaceutics12010059

**Published:** 2020-01-10

**Authors:** Eliška Böhmová, Robert Pola, Michal Pechar, Jozef Parnica, Daniela Machová, Olga Janoušková, Tomáš Etrych

**Affiliations:** Institute of Macromolecular Chemistry, Czech Academy of Sciences, Heyrovsky Sq. 2, 162 06 Prague 6, Czech Republic; pola@imc.cas.cz (R.P.); pechar@imc.cas.cz (M.P.); jozef.parnica@gmail.com (J.P.); danielamachova.chemi@gmail.com (D.M.); janouskova@imc.cas.cz (O.J.); etrych@imc.cas.cz (T.E.)

**Keywords:** cell-penetrating peptides, polymer carriers, delivery systems, HPMA copolymers, cancerostatics, diagnostics

## Abstract

Cell-penetrating peptides (CPPs) are commonly used substances enhancing the cellular uptake of various cargoes that do not easily cross the cellular membrane. CPPs can be either covalently bound directly to the cargo or they can be attached to a transporting system such as a polymer carrier together with the cargo. In this work, several CPP–polymer conjugates based on copolymers of *N*-(2-hydroxypropyl)methacrylamide (pHPMA) with HIV-1 Tat peptide (TAT), a minimal sequence of penetratin (PEN), IRS-tag (RYIRS), and PTD4 peptide, and the two short hydrophobic peptides VPMLK and PFVYLI were prepared and characterized. Moreover, the biological efficacy of fluorescently labeled polymer carriers decorated with various CPPs was compared. The experiments revealed that the TAT–polymer conjugate and the PEN–polymer conjugate were internalized about 40 times and 15 times more efficiently than the control polymer, respectively. Incorporation of dodeca(ethylene glycol) spacer improved the cell penetration of both studied polymer–peptide conjugates compared to the corresponding spacer-free polymer conjugates, while the shorter tetra(ethylene glycol) spacer improved only the penetration of the TAT conjugate but it did not improve the penetration of the PEN conjugate. Finally, a significantly improved cytotoxic effect of the polymer conjugate containing anticancer drug pirarubicin and TAT attached via a dodeca(ethylene glycol) was observed when compared with the analogous polymer–pirarubicin conjugate without TAT.

## 1. Introduction

Cell-penetrating peptides (CPPs) are short oligopeptides that were designed and described as moieties that are able to readily penetrate into living cells. Most commonly, they are used for delivery of certain cargoes into the cells. The cargo is usually a hydrophobic or a large molecule that does not easily get across the cell membrane alone. The number of CPPs described in literature is relatively high and it is still rising. The efficiency of the penetration depends on the particular structure of the CPP, as well as on the cell type, temperature, concentration, and duration of the particular experiment. According to the structure, CPPs may be categorized into the three main groups—positively charged; hydrophobic, and amphiphilic peptides [1]. The CPP can be directly attached to the cargo by a covalent bond [2] or both the CPP and the cargo can be attached together to suitable carriers [3] such as metal nanoparticles [4], polymers [5], or liposomes [6]. The detailed structure of each system can substantially affect the cell penetration process. 

Recently, various water-soluble and micellar polymer drug carriers have been developed and studied in detail; among them *N*-(2-hydroxypropyl)methacrylamide (HPMA) copolymers have a significant position. These materials are well-defined copolymers perfectly tolerated in the human body; they are water-soluble, non-immunogenic, biocompatible, and clearable from organism by renal filtration (up to molecular weight ~70,000 g·mol^−1^) [7]. The indisputable advantage of this polymer system is its hydrophilicity, enabling solubilization of various hydrophobic drugs after their attachment to the polymer [8]. Conjugation of the copolymer with either a fluorescent label or a radionuclide provides powerful polymer diagnostic probes for detection of malignant tumors [9]. It was discovered that HPMA-based copolymers (as well as other macromolecules) are preferentially accumulated in tumor tissue due to the leaky and permeable tumor vascular system. This phenomenon is known as the so-called Enhanced Permeability and Retention (EPR) effect [10]. Preferably, the EPR effect in combination with efficient CPPs bound to the polymer chain could enhance internalization of some anti-tumor drugs into the malignant cells [11,12]. In some therapeutic applications, the internalization of the drug into the cancer cells is a prerequisite to its activity. e.g., in photodynamic therapy, the development of singlet oxygen inside the target cell is a crucial parameter for the high efficacy of tumor cell destruction [13]; therefore, a polymer carrier bearing both CPP and a photosensitizer could substantially improve the outcome of the therapy.

In this work, an HPMA-based copolymer was used as a carrier of both a fluorescent label and a CPP as a cell uptake enhancer. The presented system can be used either directly as a diagnostic probe for tumor imaging or even for fluorescence-guided surgery; eventually, upon replacement of the fluorophore by an anticancer drug it might be utilized as a polymer therapeutic system with enhanced accumulation in solid tumors and with improved penetration into the cancer cells. 

For our comparative study we have chosen six different cell-penetrating peptides that can be divided into the following categories: (1) three positively charged peptides—the historically first described cell-penetrating peptide GRKKRRQRRR (TAT) [14,15] derived from the transactivator of transcription protein of the human immunodeficiency virus, minimal sequence of penetratin RRMKWKK (PEN) derived from third α-helix of the Antennapedia-based homeoprotein first discovered in *Drosophila* [16,17], and the peptide RYIRS (RY) [18]; (2) one amphipathic peptide—YARAAARQARA (YA) derived from TAT and called PTD4 [19]; and (3) two short hydrophobic peptides—VPMLK (VP) [20] and PFVYLI (PF) [21]. 

The main aim of the current work was to investigate to what extent these CPPs could enhance internalization of the labeled polymer conjugates into the cancer cells at a wide range of concentrations. Moreover, another goal was to select the most efficient CPPs and to study how the length of the spacer between the polymer carrier and CPP can influence the penetration efficacy of the system. Finally, the focus on cell internalization and the cytotoxic effect of the selected conjugates decorated with cytostatic drug pirarubicin is described.

## 2. Materials and Methods 

### 2.1. Chemicals

Methacryloyl chloride (MA-Cl), 2-thiazoline-2-thiol (TT), 2,2′-azobis(isobutyronitrile) (AIBN), 1-hydroxybenzotriazole (HOBt), *tert*-butyl alcohol (*t*-BuOH), *N*,*N*-dimethylacetamide (DMA), triisopropylsilane (TIPS), 3-mercaptopropionic acid (SH-acid), *N*,*N′*-diisopropylcarbodiimide (DIC), 2-cyano-2-propyl benzodithioate (DTB-AIBN), and *o*-phthalaldehyde (OPA) were purchased from Sigma-Aldrich, Prague, Czech Republic. *N*-Ethyldiisopropylamine (DIPEA), *N*,*N*-dimethylformamide (DMF), ethyl cyanoglyoxylate-2-oxime (Oxyma), Tenta Gel R RAM resin, 9-fluorenylmethoxycarbonyl-amino acids (Fmoc-AA), piperidine (Pip), (benzotriazol-1-yloxy) trispyrrolidinophosphonium hexafluorophosphate (PyBOP), trifluoroacetic acid (TFA), 1-(9-fluorenylmethyloxycarbonyl)amino-3,6,9,12,15,18,21,24,27,30,33,36-dodecaoxanonatriacontan-39-oic acid (Fmoc-Peg_12_-COOH), and 15-(9-fluorenylmethyloxycarbonyl)amino-4,7,10,13-tetraoxa-pentadecanoic acid (Fmoc-Peg_4_-COOH) were purchased from Iris Biotech, GmbH, Marktredwitz, Germany. Dyomics 633 amino derivative (Dy633) was purchased from Dyomics GmbH, Jena, Germany. 1-Aminopropan-2-ol (AMP) was purchased from Tokyo Chemical Industry Co., Ltd., Tokyo, Japan. Initiator 2,2′-azobis(4-methoxy-2,4-dimethylvaleronitrile) (V70) was purchased from FUJIFILM Wako Pure Chemical Corporation, Neuss, Germany. 5-Azidopentanoic acid (N_3_-pent-COOH) was purchased from Bachem, Bubendorf, Switzerland. Amino-1-(11,12-didehydrodibenzo[b,f]azocin-5(6H)-yl) propan-1-one (Dbco-NH_2_) was purchased from Click Chemistry Tools, Scottsdale, AZ, USA. Methanol, acetonitrile, and all other solvents were purchased from VWR International s. r. o., Stříbrná Skalice, Czech Republic. All chemicals and solvents were of analytical grade. The solvents were dried and purified by conventional procedures. All amino acids were l-configuration. Pirarubicin (Pir) was obtained from Meiji Seika Pharma Co., Ltd. (Tokyo, Japan).

### 2.2. Analytical Methods

Control of peptide purity and monitoring of subsequent coupling reactions with polymer carrier was performed by HPLC (Shimadzu, Kyoto, Japan) using Chromolith Performance RP-18e, 100 × 4.6 mm column (Merck, Darmstadt, Germany). A linear gradient of water-acetonitrile (5–95 vol % of acetonitrile) with the presence of 0.1 vol % TFA was used as a mobile phase and the flow rate was set to 5 mL·min^−1^. Detection was performed via a UV/VIS photodiode array detector (Shimadzu, Japan) and a fluorescence detector RF-10AXL (Shimadzu, Japan). The content of attached peptides was determined by amino acid analysis. Samples were hydrolyzed (6 M HCl, 115 °C, 16 h in sealed ampule) then modified with OPA and SH acid immediately before injection to HPLC using the same column as was described above and for fluorescence detection (excitation 229 nm, emission 450 nm) and gradient elution (10–100% of solvent B in 35 min, flow rate 1.0 mL·min^−1^) with buffers of composition as follows: solvent A: 0.05 M sodium acetate buffer, pH 6.5; solvent B: 300 mL of 0.17 M sodium acetate and 700 mL of methanol. Molecular weight and dispersity were determined by size exclusion chromatography (SEC) using HPLC (Shimadzu) equipped with a TSK 3000 SWXL column (Tosoh Bioscience, Tokyo, Japan) and refractive index (RI) and UV (Shimadzu), differential viscometer ViscoStar III, and multiangle light scattering DAWN 8 EOS (Wyatt Technology Corp., Santa Barbara, CA, USA) detectors. Measurements were performed using a mobile phase composed of 80 vol % methanol and 20 vol % 0.3 M acetate buffer (pH 6.5) at a flow rate of 0.5 mL·min^−1^. Contents of reactive TT groups (ε_305_ = 10,300 L·mol^−1^·cm^−1^, MeOH), Dy633 (ε_633_ = 200,000 L·mol^−1^·cm^−1^, EtOH), and pirarubicin (ε_488_ = 11,300 L·mol^−1^·cm^−1^, MeOH) were determined spectrophotometrically using a SPECORD 205 Spectrometer (Analytik Jena AG, Jena, Germany). Matrix-assisted laser desorption/ionization time of flight mass spectroscopy (MALDI-TOF MS) was performed on a Bruker Biflex III mass spectrometer. The molecular mass of the peptide products was determined using mass spectrometry performed on an LCQ Fleet mass analyzer with electrospray ionization (ESI MS) (Thermo Fisher Scientific, Inc., Waltham, MA, USA).

### 2.3. Synthesis of Monomers

*N*-(2-hydroxypropyl)methacrylamide (HPMA) was prepared by the reaction of methacryloyl chloride with 1-aminopropan-2-ol in dichloromethane as described earlier [22]. 3-Methacrylamidopropanoic acid (Ma-β-Ala-OH) was synthesized by the reaction of methacryloyl chloride with 3-aminopropanoic acid in aqueous alkaline medium [23]. 3-(3-Methacrylamidopropanoyl)thiazolidine-2-thione (Ma-β-Ala-TT) was prepared by the reaction of Ma-β-Ala-OH with 4,5-dihydrothiazole-2-thiol in the presence of 4-dimethylaminopyridine [23]. The monomer was characterized using HPLC and ESI MS (calculated 258.3, found 259.1, M + H).

### 2.4. Synthesis of the Polymer Precursor Poly(HPMA-co-Ma-β-Ala-TT)

The title copolymer with reactive TT groups was prepared by reversible addition-fragmentation chain-transfer (RAFT) polymerization of HPMA (1.5 g, 10.47 mmol) and Ma-β-Ala-TT (369 mg, 1.42 mmol) in a mixture of 85 vol % of *t*-BuOH and 15 vol % of DMA. Reaction was carried out in a sealed ampule under argon at 40 °C for 16 h. DTB-AIBN (13.17 mg) was used as the chain-transfer agent (CTA) and V-70 (9.17 mg) was used as initiator (INI) [23]. The resulting polymer solution was precipitated into a mixture of acetone and diethylether (1:1). Polymer was then filtered and dried in a vacuum. The polymer was dissolved in DMA (1.5 mL) with 20 wt % of AIBN and heated to 80 °C for 2 h to remove the terminal DTB groups [24]. The final polymer precursor **1** was then analyzed by SEC; molecular weight was ${\bar{M}}_{W}$ = 27,500 g∙mol^−1^ with polydispersity index *Ð* = 1.03. Analysis using UV/VIS spectrophotometer showed content of TT groups equal to 11.6 mol %.

### 2.5. Synthesis of Peptides

Peptides were synthesized using a standard Fmoc strategy using a Liberty Blue microwave peptide synthesizer (CEM, Matthews, NC, USA) and Tenta Gel R ring amide resin. Starting from the C-terminus the synthesis was performed automatically with 2.5 equivalent of the *N*-Fmoc-protected amino acid derivative, 2.5 equivalent of DIC as an activator, and 2.5 equivalent of Oxyma as an activator base in DMF in each step. After microwave coupling and washing, Fmoc protecting group was cleaved. Part of the final peptide was than modified with Fmoc-Peg_12_-COOH or Fmoc-Peg_4_-COOH, respectively. In the last coupling step, N_3_-pent-COOH was attached to the N terminus. Finally, the peptides were cleaved from the resin using a mixture of 95 vol % TFA, 2.5 vol % TIPS, and 2.5 vol % water for 3 h. The resin was removed by filtration, the filtrate was concentrated under reduced pressure, and the crude peptide was isolated by precipitation with cold diethyl ether followed by filtration. Purity and identity of the prepared peptides was determined by HPLC and MALDI-TOF MS.

### 2.6. Synthesis of Enzymatically Cleavable Pirarubicin Derivative—N_3_-pent-GFLG-Pir 

N_3_-pent-GFLG-OH was prepared on 2-chlorotrityl chloride resin as described earlier [23]. Pirarubicin was attached to the peptide carboxylic group using DIC/NHS activation similarly as described in our previous work [23], except that the condensation of the peptide with pirarubicin was performed without DIPEA. Finally, the product was purified using column chromatography (silicagel, chloroform, and methanol 95:5). The yield was 27.5 mg (0.0244 mmol, 51%) of the title compound. ESI MS (calculated 1126.5, found 1149.5 M + Na).

### 2.7. Synthesis of the Polymer Conjugates

Fluorescently labeled polymer–peptide conjugates were synthesized via a three-step procedure. First, polymer precursor **1** (120 mg, 91.7 μmol TT) was reacted with Dbco-NH_2_ (2 mol %, 4.37 mg, 15.8 μmol) in DMA (1.2 mL) in the presence of DIPEA (2.7 μL) as shown in Scheme 1. Monitoring of the coupling was done by HPLC. After 1 h, all Dbco-NH_2_ was reacted. In the second step, fluorescent dye Dy633-NH_2_ (0.5 mol %, 3 mg, 3.89 μmol) dissolved in DMA (100 μL) was added in the presence of DIPEA (0.67 μL) to the reaction mixture. To remove the remaining TT groups, AMP (7 μL, 91.7 μmol) was added to the reaction mixture. The resulting polymer precursor **2** was separated by precipitation to mixture of acetone and diethylether (1:1). The third step of synthesis was copper-free click reaction of azide derivatives of peptide TAT, PEN, YA, RY, VP, and PF (1.5 mol %) with Dbco groups of the polymer. After 1 h, all peptide was attached according to HPLC analysis. Final polymer–peptide conjugates were isolated by precipitation to a mixture of acetone and diethyl ether (1:1) and dried to constant weight. Reference polymer **REF** was prepared similarly. Briefly, polymer precursor **1** (20 mg, 15.3 μmol TT) was reacted with Dy633-NH_2_ (0.5 mol %, 0.5 mg) in the presence of DIPEA (0.1 μL) in DMA (0.2 mL). Remaining TT groups were removed by addition of AMP (1.2 μL, 15.3 μmol) and polymer was precipitated and dried, monitoring of all reaction was performed by HPLC. All polymers were dissolved in water, purified by chromatography on Sephadex G 25 resin in water (PD 10 column, GE Health care), and freeze-dried. The amount of fluorescent dye was determined spectrophotometrically in methanol (ε_633_ = 200,000 L·mol^−1^·cm^−1^) and amount of bound CPP was determined by amino acid analysis. For preparation of polymer–peptide conjugates with various spacer lengths, polymer precursor **2** was reacted with N_3_-pent-TAT/PEN and N_3_-pent-Peg_12_-TAT/PEN peptides, respectively. 

Synthesis of drug bearing polymer–peptide conjugate was performed in three steps. In the first step, polymer precursor **1** (25 mg, 19.1 μmol TT) was reacted with Dbco-NH_2_ (6 mol %, 2.63 mg, 9.5 μmol) in DMA (0.3 mL) in the presence of DIPEA (1.6 μL). Remaining TT groups were removed by adding AMP (1.55 μL, 20.3 μmol) to the reaction mixture. The resulting polymer precursor was precipitated to a mixture of acetone and diethyl ether (1:1) and dried. The second step of synthesis was 1 h lasting “click” reaction of N_3_-pent-Peg_12_-TAT (4 mg, 0.944 μmol) to the Dbco-containing polymer precursor in water (0.5 mL). The reaction mixture was freeze-dried and the solid was dissolved in DMA (0.3 mL). Finally, N_3_-pent-GFLG-Pir (4.78 mg, 4.24 μmol) was bound to the polymer precursor by “click” reaction. All reaction steps were monitored by HPLC. The final product **P-T12-Pir** was isolated by precipitation to ethyl acetate. The content of pirarubicin in the polymer conjugate was determined spectrophotometrically in methanol (ε_488_ = 11,300 L·mol^−1^·cm^−1^) and the content of CPP was determined by amino-acid analysis. The conjugate without peptide **P-Pir** was prepared by the same procedure, only the amount of Dbco-NH_2_ was reduced (3.5 mol %, 1.54 mg, 5.53 μmol) and the step with the peptide click reaction was excluded. N_3_-pent-GFLG-Pir (3.98 mg, 3.53 μmol) was bound to polymer precursor as described above.

Characteristics of all prepared polymers are summarized in Table 1.

### 2.8. Cell Culture

The HeLa cell line (LGC Standards, Poland) was cultured in Dulbecco’s modified Eagle medium (DMEM) supplemented with 100 U of penicillin, 100 μg⋅mL^−1^ streptomycin, and 10% fetal bovine serum in 25 cm^2^ flasks. Cells were cultivated in a humidified incubator at 37 °C with 5% CO_2_. The chemicals were purchased from Life Technologies/Gibco, Prague, Czech Republic.

### 2.9. Flow Cytometry

The HeLa cell line was incubated with fluorescently labeled polymer–peptide conjugates for 1 h at 4 °C and 37 °C. For measurement at 4 °C, HeLa cells were harvested with 0.05% trypsin (Thermo Fisher Scientific, Czech Republic) at 37 °C for 3 min. Subsequently, cells were washed with 0.5% bovine serum albumin in phosphate buffered saline (0.01 M) (BSA/PBS), pre-cooled for 15 min and incubated with fluorescently labeled polymer–peptide conjugates in the dark for 1 h at 4 °C. After incubation, the cells were washed once with BSA/PBS, centrifuged at 1500 rpm for 3 min and re-suspended in 1 μM Sytox Blue Dead Cell Stain (Thermo Fisher Scientific, Czech Republic) to distinguish live and dead cells. For measurement at 37 °C, HeLa cells were seeded 24 h prior to the experiment on 24-well plates at a density 1.5 × 10^5^ cells per well. Polymer–peptide conjugates were then added together with fresh DMEM and incubated at 37 °C in 5% CO_2_ and humidified atmosphere. Subsequently, cells were washed with BSA/PBS, harvested with 0.05% trypsin at 37 °C for 3 min, transferred to tubes, centrifuged at 1500 rpm for 3 min, and re-suspended in PBS/BSA containing 1 μM Sytox Blue Dead Cell Stain (Thermo Fisher Scientific, Czech Republic). All measurements were acquired using a FACSVerse™ flow cytometer (Becton Dickinson Czechia, s.r.o., Prague, Czech Republic) and analyzed by FlowJo software version 10 (Tree Star Inc., Ashland, OR, USA); the median of fluorescence intensity was determined. The concentration of polymer conjugates was adjusted to the value corresponding to a final concentration of peptide 1000 μM. This stock solution was diluted with PBS to obtain all used concentrations. **REF** polymer conjugate was used in a final concentration 11.6 mg⋅mL^−1^, which corresponds to the amount of dye in the corresponding polymer–peptide conjugates. HeLa cells without a polymer were used as a negative control. All samples were measured in duplicates in two independent experiments.

### 2.10. Laser Scanning Confocal Microscopy

Laser scanning confocal microscopy (LSCM) was used to visualize the intracellular uptake of selected polymer conjugates. HeLa cells were seeded 24 h prior to an experiment (1.5 × 10^5^ cells per well) to confocal chambers (1 μ-dishes providing a 35 mm^2^ growth area with 500 μL cell culture medium) with four chambers in a 20 mm micro-well and a #1 cover glass (0.13–0.16 mm) (Bio-Port Europe, Prague, Czech Republic). Medium was replaced by fresh medium with dissolved polymer–peptide conjugates and added to confocal chamber and incubated either at 4 °C or at 37 °C for 1 h. After the incubation time, cells were washed twice with PBS and measured by confocal microscopy. Hoechst 33342 was used for visualization of cell nuclei (5 μg·mL^−1^, Thermo Fischer Scientific, Czech Republic) and cell membranes were stained with CellMask™ Green (1 μg⋅mL^−1^, Thermo Fisher Scientific, Czech Republic) for 1 h before imaging at 4 °C or 10 min at 37 °C. Observation of samples was performed by Olympus IX83 coupled with FV10-ASW software (Olympus, Prague, Czech Republic). The samples were scanned using the 60× oil objective Plan ApoN (1.42 numerical apertures). Dy633-labeled polymers were excited at 637 nm and emitted light was detected through a 650–750 nm filter. For the detection of Hoechst 33342 dye-stained nuclei, an excitation wavelength of 405 nm was used and emitted light was detected through a 420–500 nm filter. CellMask Green-stained cell membranes were excited at 488 nm and the emitted light was detected through a 500–600 nm filter. All samples were measured in duplicate in two independent experiments.

### 2.11. Cell Viability

Cytotoxicity of polymer–peptide conjugates was determined using Alamar Blue^®^ cell viability reagent (Thermo Fischer Scientific, Czech Republic) in HeLa cells according to the manufacturer’s protocol. Cells were seeded into 96-well plates in 100 μL of DMEM 24 h prior to experiment at a density of 5 × 10^3^ cells per well. The medium was then replaced by 100 μL of the fresh medium with a serial dilution of **TAT4**, **PEN4**, **REF**, and **P-T12-Pir**, **P-Pir**, respectively. Cells were subsequently incubated for 24 h in 5% CO_2_ at 37 °C. Then the medium was replaced by 90 μL of fresh medium with 10 μL of Alamar Blue^®^ reagent and incubated for the next 4 h in 5% CO_2_ at 37 °C. Resazurin, the active compound of Alamar Blue^®^ reagent, was reduced to the highly fluorescent compound resorufin only in viable cells. The fluorescence intensity was measured using a Synergy Neo plate reader (Bio-Tek, Prague, Czech Republic) with excitation at 550 nm and emission at 590 nm. Non-treated cells were used as a negative control and free pirarubicin as a positive control. All samples were measured in triplicate in three independent experiments. The half maximal inhibitory concentration value IC_50_ was expressed as a concentration of CPP ± SD and pirarubicin ± SD, respectively. The cytotoxic response of HeLa cells to the above-mentioned polymer–peptide conjugates was measured in the concentration range from 98 nM to 100 μM of CPP, concentration of polymer conjugate **REF** was used in a range from 1.1 mg⋅mL^−1^ to 1.1 μg⋅mL^−1^. In the case of drug bearing polymer conjugates, the concentration varied from 43.2 nM to 44.3 μM of pirarubicin.

### 2.12. Statistical Analysis

Results were plotted as average ± standard deviation (SD). All statistical analysis was performed using GraphPad Prism 5.03. An ANOVA test was followed by Dunnett’s or Turkey’s test. A value of *** *p* < 0.001, ** *p* < 0.01, and * *p* < 0.05 was considered statistically significant. All IC_50_ (the concentrations of the CPP or drug reducing the cell viability to one half) values were obtained from calculations corresponding to fitting by logistic S-curves.

## 3. Results and Discussion

### 3.1. Synthesis and Characterization of Polymer Precursor and Polymer–Peptide Conjugates

The course of synthesis is presented in Scheme 1. Reactive polymer precursor **1** was prepared via a controlled radical RAFT copolymerization providing copolymers with a very narrow distribution of molecular weights even in subsequent reaction steps. The low dispersity of the copolymers is a very important feature of the biomaterial intended for biomedical applications. It will result in a more uniform pharmacokinetic behavior that will be advantageous, especially in the case of tumor targeting. Last but not least, the polymer therapeutics or diagnostics with low dispersity will probably receive eventual regulatory approval for clinical applications more easily than those with a broad molecular weight distribution. 

Fluorescently-labeled polymer precursor **2** was synthesized by the reaction of TT groups of the precursor **1** with the amino derivative of Dbco followed with the amino-modified red-excited fluorescent dye Dyomics 633, both in the presence of DIPEA. Unreacted TT groups were removed by addition of excess of AMP. Polymer–peptide conjugates were synthesized by copper-free click reaction of polymer **2** and the respective peptide. The polymer conjugates with pirarubicin (**P-Pir** and **P-T12-Pir**) were prepared analogically by click reaction of N_3_-pent-GFLG-Pir for **P-Pir** or N_3_-pent-GFLG-Pir and N_3_-pent-Peg_12_-TAT for **P-T12-Pir**, respectively, to polymer precursor containing Dbco groups. Scheme of prepared conjugate **P-T12-Pir** is shown in Figure A1 (Appendix A). Physicochemical characteristics of the prepared samples are listed in Table 1. Weight average molecular weights ${\bar{M}}_{w}$ and dispersity *Ð* of copolymers and conjugates were determined using the SEC equipped with light-scattering detector. Due to the interference of the fluorescence of Dye with the scattering detector, ${\bar{M}}_{w}$ and *Ð* was estimated from RI detector data. No significant change in ${\bar{M}}_{w}$ and *Ð* was found during the synthesis of CPP-containing fluorescently labeled or drug decorated polymer biomaterials, see Figure A2 in Appendix A. The molecular weight of all polymers was around 30,000 g/mol. All the CPPs were attached to the HPMA-based fluorescently labeled polymer precursor using the same polymer–peptide molar ratio, reaching approximately 1.5 mol %. To compare the effectivity of the respective systems, a control polymer without any penetrating peptide was also synthesized and evaluated to assess the eventual influence of the dye and of the polymer carrier to the penetration activity of the polymer conjugates.

We can summarize that polymer–peptide conjugates with various CPPs were successfully prepared and characterized. The size exclusion chromatography verified that the molecular weights and dispersities of polymer–peptide conjugates were not significantly affected by the attachment of the peptides. 

### 3.2. Cell Penetration Ability of Polymer–Peptide Conjugates

Cell-penetrating peptides are described as moieties that are able to increase cellular uptake of a cargo in most cell lines [25]. We have selected a well-known cancer cell line HeLa (human cervix epitheloid carcinoma cells) as a representative cell line for biological experiments. Polymer–peptide conjugates containing in average three molecules of the peptide per polymer chain (1.5 mol %) were incubated with HeLa cells. The cell penetration ability of the conjugates listed in Table 1 was tested both at 37 °C corresponding to human body temperature and at 4 °C (with precooled cells and solutions) under the conditions excluding any active transport through the cell membrane via endocytosis (Figure 1) [26]. 

The uptake of the polymer–peptide conjugates incubated with the cells was measured using flow cytometry. The observed fluorescence signal was related to the fluorescence signal of the control polymer (**REF**) without any CPP. Consequently, Figure 1 shows the ratio of the cell penetration between respective conjugates and control polymer **REF** expressed by median of the cell-associated fluorescence intensity.

It is evident (Figure 1A) that polymer conjugates with peptides VPMLK, YARAAARQARA, and RYIRS do not penetrate cells better than the control polymer at any of the concentrations used at 4 °C (green bars) nor at 37 °C (orange bars) when the endocytosis contributes to the uptake of the macromolecules. In contrast to these inactive conjugates, the polymer conjugate decorated with a short hydrophobic peptide PFVYLI was found at 37 °C inside the cells at about 4 to 5 times higher concentration than the control polymer **REF**, thus the endocytosis was present as a dominant mechanism of uptake at concentrations 1 μM and 10 μM. At 4 °C, the increase in uptake was much less pronounced, showing up to 2 times higher concentration than that determined for REF. At the highest concentration, the peptide conjugate was only 1.8 times better than the control polymer at both 4 and 37 °C. 

It is obvious (see Figure 1B) that peptides TAT and PEN contribute in various degrees to the increased cellular uptake of the corresponding polymer conjugates. To demonstrate that TAT and PEN act as CPP, the polymer–peptide conjugates were incubated with HeLa cells at 4 °C. In particular, polymer conjugate **TAT4** exhibited almost seven times higher cell penetration level compared with **REF** already at concentration 1 μM. This effect steadily grew with increasing concentration of the peptide up to concentration 50 μM of CPP. At this concentration, the cell uptake was more than 53 higher than that of the control polymer. Polymer conjugate **PEN4** at 4 °C at concentrations of 1 and 10 μM even slightly outperformed **TAT4**; however, the difference was non-significant. At higher concentrations (25 and 50 μM) at 4 °C, **PEN4** did not exhibit any more improvement; a kind of plateau was reached. In contrast to this, **TAT4** penetration (compared with **REF**) was continuously growing with increasing concentration. 

At physiological temperature 37 °C, conjugate **TAT4** exhibited 14 times enhanced cellular uptake (expressed by increase of relative fluorescence intensity) compared to **REF** already at concentration 0.1 μM. The relative uptake of the polymer conjugate was increasing up to concentration 10 μM; it virtually went down at higher concentrations. This observation can be explained by the linear increase of the absolute fluorescence intensity of the reference polymer **REF** compared to the non-linear increase of the fluorescence intensity of both polymer–peptide conjugates (see Figure 2). This data shows that CPP-conjugates are internalized more than polymers conjugates without any CPP. We supposed that very high levels of uptake of CPP-bearing conjugates is probably related to the simultaneous influx of this conjugate into cells by endocytosis and penetration.

Conjugate **PEN4** exhibited similar concentration dependence of the penetration ability; however, the relative increase of the fluorescence intensity compared to **REF** was lower than in the case of **TAT4** for all measured concentrations.

The viability of the cells was detected with nucleic acid staining Sytox Blue entering only dead cells using flow cytometry. The fluorescence intensity values around 10^2^ correspond to the live cell population (Figure A3A,B in Appendix A); the values around 10^4^ belong to the dead cells (Figure A3B in Appendix A). An example of the gating of HeLa cells is shown on Figure A3C in Appendix A. Region of interest of live HeLa cell is highlighted. This region was estimated by Sytox Blue Dead Cell Stain. In Figure A3D in Appendix A, an example of the fluorescence intensity histograms after incubation of polymer–peptide conjugates (**REF**—blue, **TAT4**—green, and **PEN4**—orange) at the concentration 10 μM at 37 °C with HeLa cells is shown. The red histogram belongs to non-treated HeLa cells. 

As we already mentioned, the peptide-free polymer with just a fluorescent label (**REF**) was used as a negative control. The non-specific cell binding (and/or internalization) was increasingly linear with increasing concentration of the sample (purple dots on Figure 2). The linear dependence was observed for both 4 and 37 °C though with different slopes. This linear dependence was studied up to concentration 100 μM; at higher values the copolymer **REF** becomes insoluble in PBS. Contrary to **REF,** the polymer–peptide conjugates **TAT4** (orange dots on Figure 2) and **PEN4** (green dots on Figure 2) showed some kind of dependency of Median Fluorescence Intensity (MFI) on concentration. We hypothesize that the decrease of the fluorescence with increasing concentration is based on the interaction of the fluorophore on the CPP-containing conjugates located on the cell membrane or internalized into the cells with other molecules located in the cell membrane or cytosol. These molecules could interact with the fluorophores and are due to the non-radiant energy transition. Consequently, it led to the observed decrease of the relative penetration efficacy (related to **REF**) at the highest concentration shown in Figure 1B.

The flow cytometry study revealed that from several chosen CPPs only TAT, PEN, and marginally PF peptides are able to significantly increase cellular uptake of fluorescently labeled HPMA-based copolymers. Unfortunately, the difference between the control and the **PF4** polymer–peptide conjugate was quite low in comparison with following **TAT4** and **PEN4** conjugates. This finding led us to the decision to focus only on these two peptides in our further investigations.

### 3.3. Spacer-Length Impact on Cellular Localization

Hydrophilic synthetic polymers (including HPMA-based copolymers) generally adopt in aqueous solution a random coil conformation. We hypothesized that the insertion of a short poly(ethylene glycol) spacer between the polymer chain and the CPP might help to expose the peptide (that could be buried inside the polymer coil) and thus improve the interaction of the peptide with the cell membrane. The results of the corresponding flow cytometry experiments are summarized in Figure 3. Again, the median of the fluorescence intensity of the polymer conjugates is related to the control polymer **REF** (with fluorescence intensity equal to 1).

In the case of TAT–peptide conjugates (Figure 3A), the relative fluorescence intensities (corresponding to the relative penetration efficacy) increase with increasing length of the spacer for concentration 10 μM and higher. At concentration 50 μM, conjugate **TAT12** exhibits 2.8 times higher penetration efficacy than conjugate **TAT0**. PEN conjugates (Figure 3B) show a similar trend; however, the efficacy of the chain penetration is not proportional to the spacer length. The tetra(ethylene glycol) spacer of **PEN4** is not long enough to improve the penetration compared with spacer-free conjugate **PEN0**; only the longest spacer with 12 ethylene glycol units leads to significant penetration improvement of the corresponding conjugate **PEN12**. This effect is most pronounced at concentration 25 μM where **PEN12** conjugate with the long spacer has two times higher penetration efficacy than the shorter analog **PEN4**. We can only speculate that the possible reason for the different spacer length influence between the PEN and TAT conjugates lies in the different peptide structures affecting the non-covalent interactions with its neighborhood, including the cell membrane.

We have proved that the cellular uptake of fluorescently labeled polymer–peptide conjugates is highly dependent on the length of the oligoethylene glycol spacer between the TAT or PEN peptide and the polymer chain. The resume from this observation is indisputably the longer, the better.

### 3.4. Confocal Microscopy Imaging

A confocal microscopy study was performed to support and extend the data obtained by flow cytometry. Confocal microscopy imaging was used to map localization of CPP conjugates. The polymer conjugates with the highest cell penetration activity, **TAT12** and **PEN12**, were selected for the confocal microscopy experiments. Conjugate **REF** was used as a control. The polymers were incubated with HeLa cells for 1, 2, or 4 h at 4 or 37 °C.

CellMaskGreen (green) and Hoechst33258 (blue) were used for staining of the cell membranes and nuclei, respectively (Figure 4A). The cells incubated with polymer conjugate **TAT12** containing Dyomics 633 as a fluorophore are depicted in Figure 4B. Figure 4C represents the merged fluorescent image of the cell staining with the fluorescent polymer **TAT12** revealing the intracellular localization of the polymer conjugate after 2 h.

Cellular distribution of CPP–polymer conjugates on the cell membranes and inside the cells after various incubation times is shown in Figure 5. At physiological temperature (Figure 5A), conjugate **TAT12** was localized in cytosol already after 1 h and the concentration of the polymer in cytosol was increasing in time, which is in agreement with the flow cytometry data. Although TAT peptide is described in the literature as a nuclear localization sequence [27], the presence of the polymer in the cell nuclei was negligible. It is probably caused by the large molecular weight of the respective conjugate. In the case of **PEN12** conjugate, we observed lower cell internalization compared with **TAT12**. In contrast with the CPP conjugates, polymer **REF** was present in the cells in the lowest concentration. The limited amount **REF** was slowly growing due to fluid-phase endocytosis after longer incubation times.

The same experiments performed at 4 °C (Figure 5B) revealed that endocytosis is not the only process responsible for the internalization of the CPP conjugates; a direct penetration of the cell membrane is another mechanism of cell entry. The cells were cooled to 4 °C for 15 min prior to the addition of the conjugates to rule out any possible endocytosis. The images demonstrate that the rate of internalization of all polymer conjugates is dramatically lower than at physiological temperature. While conjugate **REF** was not present in cells at all, even after 4 h, the CPP conjugates bind to the cell membrane and TAT12-containing conjugate is partially localized even in the cytosol of the cells, although the majority of the fluorescence comes from the cell membrane (see red signal on Figure 5B upper right image). However, the detectable amount in the cytosol was found only for this conjugate at higher magnification. Unfortunately, longer incubation time than 4 h at the low temperature led to a significant decrease of cell viability. 

We can sum up that the confocal microscopy images clearly proved the significant CPP-activity of TAT12 as cellular localization of the polymer–peptide conjugates at 4 and 37 °C in various time points was observed. The enhanced cellular localization for PEN12 containing polymer–peptide conjugate was observed mainly at 37 °C, it was much less pronounced at 4 °C. 

### 3.5. Cell Viability Results

The cell viability dependence on the concentration of the fluorescently labeled polymer conjugates after 24 h incubation is demonstrated in Figure 6A. The data indicate that the increasing polymer concentration has some cytotoxic effect, which cannot be attributed to the CPP alone as a similar trend can be observed also for the control polymer **REF** without any peptide. IC_50_ values (the concentrations of the CPP reducing the cell viability to one half) determined from the corresponding curves were 95.3 ± 2.6 μM, 82.9 ± 4.8 μM, and 216.5 ± 3.6 μM for **TAT4**, **PEN4**, and **REF**, respectively. In the case of polymer **REF**, the virtual molar concentration of the absent CPP refers to the molar concentration of CPP in the corresponding polymer–peptide conjugates of the same weight concentration. It is evident that the cytotoxicity of the polymers at the concentrations above 2 μM of CPP is at least partially caused by the fluorophore Dy-633. Nevertheless, the CPP concentrations about 100 μM (corresponding to approximately 1.2 mg/mL of the polymer) and above are about two orders of magnitude higher than possible intracellular concentrations would be after eventual in vivo application of the similar polymer conjugates as potential therapeutics or diagnostics.

Figure 6B demonstrates analogous viability-concentration dependence for polymer conjugates **P-T12-Pir** and **P-Pir** containing a cancerostatic drug pirarubicin. Free pirarubicin in a concentration range from 9.8 ng·mL^−1^ to 100 μg·mL^−1^ was used as a positive control. The highest used concentration of the **P-T12-Pir** conjugate corresponds to 50 μM of CPP. IC_50_ values (the concentrations of the Pir reducing the cell viability to one half) determined from the corresponding curves were 18.3 ± 6.2 μg·mL^−1^; 41.3 ± 3.8 μg·mL^−1^ and 1.03 ± 0.3 μg·mL^−1^ for **P-T12-Pir**, **P-Pir**, and free **Pir**, respectively. Finally, the IC_50_ value of TAT-bearing conjugate was more than two times lower compared with **P-Pir** conjugate without any CPP. At the same time, the concentration of CPP present on **P-T12-Pir** at IC_50_ value is approximately 20.7 μM. The IC_50_ of free pirarubicin was one order of magnitude lower. This observation is typical when comparing cytotoxicity of free and polymer bound drug [23,28,29]. The positive effect of CPP is more pronounced after relatively short incubation time (24 h) with the cells due to the faster internalization of CPP–polymer conjugates compared with the peptide-free conjugates. Interestingly, after longer incubation time (72 h) the cytotoxicity of both **P-T12-Pir** and **P-Pir** become the same (11.1 ± 1.6 μg·mL^−1^); thus, we conclude that the effect of CPP decreases in time and the amount of the internalized polymer conjugates is similar regardless of the presence or absence of CPP in the polymer conjugate. 

We can conclude that the cytotoxic effect of pirarubicin bearing polymer conjugate was more than two times improved by the incorporation of the cell-penetrating peptide TAT with the dodeca(ethylene glycol) spacer after 24 h of incubation with HeLa cells.

## 4. Conclusions

The major goal of this study was to investigate the influence of a CPP covalently attached to water-soluble polymer–drug conjugate onto the cellular uptake of the conjugate and consequently also its cytotoxicity. Various CPP and spacers between polymer chain and CPP were studied in detail and compared. Fluorescently labeled polymer conjugates decorated with cell-penetrating peptides often mentioned in literature were successfully prepared using copper-free click chemistry. Cell-penetration ability was compared considering identical molar amount of peptide on each conjugate representing in average three molecules of peptide per one polymer chain. Flow cytometry showed that TAT peptide attached along the polymer chain is rightfully the most favorite and efficient in delivery of cargo across the cell membrane. At lower concentrations, a minimal sequence of penetratin (PEN) also showed decent cell-penetrating ability. This observation was supported also by confocal microscopy experiments. 

Concerning the spacer length influence, the longest dodeca(ethylene glycol) spacer proved to have a favorable effect to the cell penetration of the both studied polymer–peptide conjugates compared to the corresponding spacer-free polymer conjugates while the shorter tetra(ethylene glycol) spacer improved only the penetration of the TAT conjugate, but it did not improve the penetration of the PEN conjugate.

After attachment of a cytostatic drug pirarubicin, we could observe significantly increased cytotoxicity of the polymer–drug-CPP conjugate compared with the polymer–drug conjugate without CPP. The improved cytotoxicity effect is apparently caused by TAT-mediated enhanced penetration into HeLa cells. 

Efficient internalization of a cytostatic drug into the target malignant cells is often a prerequisite for successful therapy; e.g., for application of DNA intercalators or photosensitizers producing singlet oxygen in photodynamic therapy, the intracellular localization of the drug is an absolute necessity. We believe that the CPP–polymer conjugates with various cytostatic drugs could be used as a new generation of therapeutic agents for a more efficient cancer treatment. Finally, the presented CPP–polymer conjugates with fluorophores could eventually be used as polymer probes for diagnosis or for fluorescently guided surgery of various malignant tumors. The efficacy of the presented TAT- and PEN-decorated polymer conjugate will be evaluated in the oncoming study for its in vivo efficacy with the aim of studying in detail the combination of EPR effect and CPP penetration in tumor models.

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
