# Peer review of "Polymer Cancerostatics Containing Cell-Penetrating Peptides: Internalization Efficacy Depends on Peptide Type and Spacer Length"

_pharmaceutics, 2020, doi:10.3390/pharmaceutics12010059_

Round 1

Reviewer 1 Report

The authors in work entitled: “Polymer cancerostatics and diagnostics containing cell-penetrating peptides: Internalization efficacy depends on peptide sequence and spacer length“ demonstrated  interaction of newly prepared peptide polymer conjugates. The internalization and ability of drug transport was evaluated in HeLa cells. The promising candidates were selected. The present manuscript is well organized and consisted of new results. However, some comments should be addressed.

- I suggest to clearly refer to appendix figures, e.g. Figure A2 is not clear where the reader should find it. Figure A1 is not mentioned in the text.

- p. 8 Fig. 1 and 2 should be Figure 1 and 2

- p.10 r.351 Fig.2. Should be Figure 2

- The resolution of Figure 1 should be higher, mainly B-E parts.

- P. 11 r.391 Similarly as before Figure A3 should refer to appendix. Otherwise it is a bit confusing for the readers to follow the text if the authors mean 3A or A3. Maybe the nomenclature of appendixes figures could be changed.

- P.13 r.468 The Authors mentioned that “the direct penetration of the cell membrane is another mechanism of the cell entry”. What is the actual size of the CPP conjugates? How can these conjugates penetrate into the cells? The authors suggest this statement from Figure 5B. However, in my opinion the localization of the TAT12 is more likely in plasma membrane. It is difficult to see cytoplasmic localization as it was in Figure 5A. Maybe the process is only the adsorption of the conjugates on the plasma membrane. Could the authors present better representation of this image with cytosolic localization of the TAT12? The discussion should be supported with the appropriated references.

- Figure 6 is divided in to A and B parts, which are missing in Figures.

- The discussion part has lack of information, e.g. a comparison of the obtained results with another reported studies. What is novelty and the advantage of the proposed approach. Maybe also subtitles should refer to main highlight of the subsections.

- An important review in this field was recently done by Silva, Almeida and Vale. (Biomolecules. 2019 Jan; 9(1): 22. Combination of Cell-Penetrating Peptides with Nanoparticles for Therapeutic Application: A Review ). It can be helpful for the interpretation of the results.

- The conclusion part does not represent compact section. The main highlight of the manuscript should be summarized.  Future research directions may also be mentioned.

Author Response

Please see attached document with point-by-point response to Reviewer 1 comments.

Reviewer 2 Report

This manuscript details experiments that aim to underline the potential of cell penetrating peptides (CPP) to improve the uptake of anticancer drugs.  This is a hot topic research field with many advances recently been published. The main findings of the presented work are: 1) HPMA can elegantly been linked with CPPs using copper-free click chemistry. 2)  4 or 12 PEG spacers can dramatically improve the cellular uptake of HPMA linked with CPPs TAT and PEN. 3) The cytotoxic effect of the anti-cancer drug pirarubicin conjugated to HPMA increases when the HPMA is linked with 12PEG-TAT

While I find these key findings worthwhile publishing, I have the following remarks and questions:

In my opinion, the title is not covered by the presented data. Polymer cancerostatics and diagnostics containing cell-penetrating peptides: Internalization efficacy depends on peptide sequence and spacer length. There is no mention of diagnostics until the last sentence, were you suggest the future applications. Secondly, you technically do not show that the sequence is important. You compared some peptides with various lengths and various AA sequences.  However, TAT contains a lot of Arginine and Lysine and works because of the positive charge. If you scramble the TAT sequence does the peptide loses its penetration ability?

In the abstract. Line 25-28: rephrase. There seems to be a word missing.

Line 306: The size exclusion chromatography verified that the molecular weightand dispersitiesy of polymer-peptide conjugates in WERE not significantly affected by the attachment of the peptides.

Figure 1. / Figure 2. I suggest merging Figure 1 A and Figure 2.  This way the reader can compare the uptake of all peptide-polymers in one figure.  In your manuscript, you first describe Figure 1A and discuss the differences, then you say it doesn`t matter, because look at Figure 2.  You make that conclusion (line 348-350) before the Figure 2.    In my opinion Figure 1B, C, D, E is supplementary information. You only describe these figures on line 381 (too far).  There also seems to be no data or conclusion on viability in that paragraph.

Figure 2 is missing the Y-axis info Figure A3 I would add to the main document because you spend a full paragraph to discuss. Line 390: increasingly linear. Conclusion of line 436: The longer the PEG spacer, the better. You follow that conclusion when testing the pirarubicin. However, for the cytotoxicity of the TAT-polymer or PEN-polymer you use the 4 PEG linker. I would like to see the cytotoxicity of the TAT12 and the PEN12.

Line 507; We already know that after linger incubation time (72h, data not shown) the effect of CPP decreases.  Please do not assume your reader already knows and please show me the different time points, including 72h or don`t mention it.

Figure 6B (the letter is missing in the figure). Please mention that the increased cytotoxicity of P-T12-Pir versus P-Pir is only significant in the higher doses.

Author Response

Please see attached document with point-by-point response to Reviewer 2 comments.

Round 2

Reviewer 1 Report

The manuscript was improved. 

Before the acceptance, please, correct the appearance of Figure 2 in the appendix (A2).

Reviewer 2 Report

Thank you for the detailed responses. I accept all changes made to the manuscript. Please note that in my pdf version, the figure A2 was out of position and on top of figure A3